# Application of Optimization Techniques in the Dairy Supply Chain: A Systematic Review

**Mohit Malik** [1] , **Vijay Kumar Gahlawat** [1,*] , **Rahul S Mor** [2,*] , **Vijay Dahiya** [3] **and Mukheshwar Yadav** [4]

1  Department of Basic and Applied Sciences, National Institute of Food Technology Entrepreneurship and Management, Kundli, Sonepat 131028, India
2  Department of Food Engineering, National Institute of Food Technology Entrepreneurship and Management, Kundli, Sonepat 131028, India
3  Department of Business Administration, Maharaja Surajmal Institute, C-4, Janakpuri, New Delhi 110058, India
4  Department of Mechanical Engineering, CUIET, Chitkara University, Rajpura 140401, India
*  Correspondence: drvijay.niftem@gmail.com (V.K.G.); dr.rahulmor@gmail.com (R.S.M.)

**Abstract:** *Background*: The global dairy market is experiencing a massive transition as dairy farming has recently undergone modernization. As a result, the dairy industry needs to improve its operational efficiencies by implementing effective optimization techniques. Conventional and emerging optimization techniques have already gained momentum in the dairy industry. This study's objective was to explore the optimization techniques developed for or implemented in the dairy supply chain (DSC) and to investigate how these techniques can improve the DSC. *Methods*: A systematic review approach based on PRISMA guidelines were adopted to conduct this review. The authors used descriptive statistics for statistical analysis. *Results*: Modernization has led the dairy industry to improve its operational efficiencies by implementing the most effective optimization techniques. Researchers have used mathematical modeling-based methods and are shifting to artificial intelligence (AI) and machine learning (ML) -based approaches in the DSC. The mathematical modeling-based techniques remain dominant (56% of articles), but AI and ML-based techniques are gaining traction (used in around 44% of articles). *Conclusions*: The review findings show insight into the benefits and implications of optimization techniques in the DSC. This research shows how optimization techniques are associated with every phase of the DSC and how new technologies have affected the supply chain.

**Keywords:** dairy industry; supply chain; optimization; machine learning; artificial intelligence; mathematical modelling





## 1. Introduction

The dairy industry worldwide is now undergoing a significant transformation. United Nations Food and Agriculture Organization dairy price index demonstrates costs 26% below the maximum in February 2014 [1]. Trade sanctions against Russia and the removal of "milk quotas" inside the EU have led to a time of oversupply and minimum pricing for milk products from China [2]. Despite this, the dairy industry is growing and is expected to reach 177 million tons of powdered milk by 2025 at an annual growth rate of 1.8%. Increasing urbanization and rising prosperity in developing economies are responsible for this rise [3]. Dairy producers in Europe have utilized intervention stocks to protect themselves against lower worldwide prices. India is the top milk producer in the world, making up 23% of the world's milk. The compound annual growth in milk production in the country is about 6.2%. In 2020–2021, milk production reached 209.96 million tons, up from 146.31 million tons in 2014–2015 [4]. The top five milk-producing states are Uttar Pradesh (14.9%), Rajasthan (14.6%), Madhya Pradesh (8.6%), Gujarat (7.6%), and Andhra Pradesh (7%) [5]. It was announced in June 2020 by the Indian government and the Department of Animal Husbandry and Dairying that there would be a $2.1 billion infrastructure

development fund. An interest subsidy scheme to help private and small businesses invest in dairy and livestock feed plants, which in turn are predicted to generate 3.5 million jobs [6]. The Ministry of Food Processing Industries has issued detailed operational scheme guidelines and started an internet-based platform for the 'Production Linked Incentive Scheme for Food Processing Industry', which has an expenditure of $1.4 billion. This money is meant to help make Indian food manufacturing companies that are world-class and help Indian brands of food products be more prevalent in international markets [7].

Strategies to improve and strengthen milk process optimization are essential in the dairy industry. The fast degradation of milk ingredients makes dairy manufacturers think cautiously about how they aim and manufacture their goods. The business model is to peek at the individuals working for the corporation [8]. They want to cut down on time and raw materials thrown away, unnecessary expenses, constraints, and errors while still making a high-quality product. In the last few years, many people have used machine learning algorithms on dairy farms and talked about how to predict many different things [9]. This article aims to find, evaluate, and synthesize the articles discussing how optimization techniques can be used to optimize the DSC. As part of artificial intelligence (AI), machine learning utilizes complicated algorithms to handle difficult situations that can't be solved with traditional methods. In dairy, machine learning is already used in many ways [10]. The dairy supply chain consists of the production and planning of milk procurement, processing, inventory, marketing, distribution, and consumer end [11]. In the DSC, optimization techniques such as mathematical modeling and AI and ML-based techniques are utilized in most phases. For example, it can detect lameness and predict the calving time by looking at motion information and action sensors. The data shows early mastitis from milking robots [12]. When implementing machine learning in the dairy industry, there isn't a good picture of the algorithms utilized, the issues solved, and the complexities that come with it [11]. The authors address optimization techniques' role in the DSC by looking at the scientific literature. The review focuses on exploring the application of optimization techniques, including ML and AI-based techniques developed for or implemented in the DSC. The authors try to investigate how these techniques can improve dairy.

This review is structured as follows; Section 2 presents the review of literature, and Section 3 describes the review methodology used to find, select, and investigate journal articles for inclusion in the study. Section 4 discusses potential dairy industry areas where optimization techniques were applied based on a review of the included articles and interpretation and discussion, along with challenges and future directions for optimizing techniques. Finally, Section 5 summarizes the findings and concludes the review.

## 2. Literature Review

Shine & Murphy [13] presented a systematic review of machine learning-based optimization techniques for dairy farms and explained how to use these techniques for various dairy farm tasks. It provides an in-depth study of the literature on optimization models for production scheduling for food production. Jouzdani & Govindan [14] discussed the sustainability aspects of the DSC in an optimization context and suggested changes from the policymakers' side to optimize the supply chain efficiently. Jachimczyk et al. [15] show the IoT evolution briefly by offering the complete architecture of how devices communicate to cloud networks to manage data. Correia et al. [16] developed a distribution model to optimize the distribution process in the DSC using the structural equation modeling technique. Jouzdani et al. [17] proposed a model to optimize the system's total costs and resolve location selection problems while keeping traffic conditions in mind. Bilgen & Çelebi [18] proposed an integrated production and distribution model to optimize each product's process.

Piramuthu & Zhou [19] presented a model to optimize the inventory for perishable products, and Gopinath et al. [20] described the optimization for lipase production. Many articles presented an integrated modeling approach to optimize decision support systems,

cost, environmental impacts, and routing problems [21–28]. Sel & Bilgen [29] review existing problems and their respective solutions to the dairy industry's quantitative modelling. Li et al. [30] overview how to optimize the complete supply chain via the traceability model and describe the total cost and profit optimization of the entire supply chain. Schuck et al. [31] review recent optimization techniques based on modeling and simulation methods. Satya & Chimakurthi [32] explores livestock production optimization and reduction of physical labor. Sel et al. [33] briefly explain the optimization of quality decay and makespan of perishable products to manage the inventory in a better way. Nemati et al. [34] explore the benefits of modeling in. operations planning and [35,36] work on the optimization of network cost and $CO_2$ emission. The researchers try to optimize the distribution process and scheduling process to predict the quantity of milk, protein, fat content, and feed intake of dairy animals [37,38]. Mor et al. [39] present a structured review of DSC practices and indicate that technological advancements can help to achieve a milestone in the direction of quality, food safety, and economic growth of the dairy sector.

Prediction of energy cost [40] and optimization of distribution along with uncertainty were explored [41]. Akbar et al. [42] examined the innovative techniques for farming and production, [43] including animal health and productivity as the main parameters for their study. Feil et al. [44] reviewed sustainability factors by indicating the major problems in an environmental, social, and economic context and also suggested solutions related to sustainable practices. In the literature review, the main application of optimization is related to cost, profit, routing problems, location problems, and decision-making. Mathematical modeling approaches like mixed integer linear programming, non-linear programming, heuristic methods, AI- and ML-based algorithms like genetic algorithms, simulation modeling, etc., were used by many authors.

## 3. Review Methodology

A systematic literature review was carried out to achieve the study goals using the "Preferred Reporting Items for Systematic Reviews and Meta-Analysis" (PRISMA) methodology, as mentioned by [45]. Very few SLR techniques in the supply chain context are focused on best practices and the unique characteristics of conducting research in the supply chain domain [46]. A systematic review is an evidence-based review of the scientific literature. By summarizing and synthesizing the available literature results, it may identify the latest condition of understanding on the study issue and aid in collecting relevant studies. Authors are motivated to conduct SLR in DSC because AI and ML research rapidly expand, especially in supply chains. This technique is relatively new in the DSC. The authors adopted a systematic review process as described by [47]. This review was conducted in five steps: developing the research question, search method, data selection, statistical analysis, and discussion [48].

### 3.1. Review Question

The study's objectives necessitated the development of research questions that would shed light on the advantages of optimization techniques in the DSC. This study area focuses on the most cutting-edge DSC techniques. The following research questions were prepared: RQ 1. Which optimization techniques are commonly used in the DSC? RQ 2. Which aspects of DSC have been studied more previously? and RQ 3. How do the Al- and ML-based optimization techniques help to improve the DSC?

### 3.2. Search Method

We devised a method for locating relevant documents. Machine learning, artificial intelligence, and DSC were the critical elements of the search string based on the study subject. Based on three themes, we came up with three subcategories: Supply chain, machine learning, artificial intelligence, and the dairy sector round out the list. Authors only used peer-reviewed articles from reliable sources to ensure the quality of their findings. Scopus and Web of Science are two important databases for peer-reviewed multidisciplinary

research papers. The authors used the Scopus and Web of Science databases to compile relevant research. Data selection is based on a set of inclusion criteria. To collect relevant articles, authors used Boolean operators OR, AND along with the following keywords ("dairy industry" OR "dairy supply chain" OR "dairy") AND ("optimization" OR "optimization techniques" OR "Artificial intelligence" OR "machine learning" OR "Mathematical Modelling"). This evaluation only included research articles from 2013 to mid-2022 since the concepts of ML, and AI were first proposed in 1952 and 1943, respectively. This evaluation needed only peer-reviewed English-language publications to ensure validity and high output standards. We searched the database using the filter function: language, year of publication, type of publication, and area of research. Table 1 lists the inclusion/exclusion parameters used to choose the articles for this review.

**Table 1.** Inclusion/exclusion criteria.

| Inclusion Criteria | Exclusion Criteria |
| --- | --- |
| Articles published from 2013–2022 | Articles published before 2013 and after 2022 |
| Published in the English language | Published in a non-English language |
| Articles focused on Optimization Techniques based on Artificial Intelligence, Machine Learning, and Dairy supply chain | Articles that are not relevant to review questions |
| Articles related to food science | Articles related to non-food domains |
| Full-text articles only | Articles that are not available in full-text form |
| Peer-reviewed research and review articles | Conference proceedings, thesis, poster abstracts, and short communications are excluded |

*3.3. Data Selection*

After scanning both databases, we found 1516 entries from Scopus and 1758 documents from Web of Science. For this review, we utilized a five-phase review process to locate the research that should be considered for inclusion. Following a search, the identification, screening, and inclusion process was carried out. From two databases, 3274 search results were discovered. 31 articles from other sources, such as back references, websites, books, and manuals, were also included. After removing 1123 duplicate entries, 2182 documents remained for further screening. Table 1's inclusion criteria were employed in the screening step to identify relevant research. During the screening process, there were two stages. There was a preliminary phase in which the abstracts and key phrases were examined. Following the first round of screening, 200 papers were selected for consideration in the second round of the selection procedure.

Following that, the full-text research papers were evaluated for eligibility. Specific reasons were given for excluding 124 articles. Several reports spoke about ML and AI but didn't use them in their study. For data extraction and analysis, 45 full-text publications were chosen. We also exclude the 31 articles from books, webpages, and manuals. After screening, the articles' fundamental features were summarized. The essential qualities demonstrate the advancement of the associated academic subject. Figure 1 depicts a flowchart for a systematic review based on PRISMA guidelines. The total number of articles published in each journal is shown in Figure 2. Only journals with several publications greater than or equal to 2 were considered in Figure 2. Figure 3 shows the year-by-year breakdown of publications. From 2013 to mid-2022, there was an upward trend in the number of published articles. Table A1 in the Appendix A presents all the articles included for review.

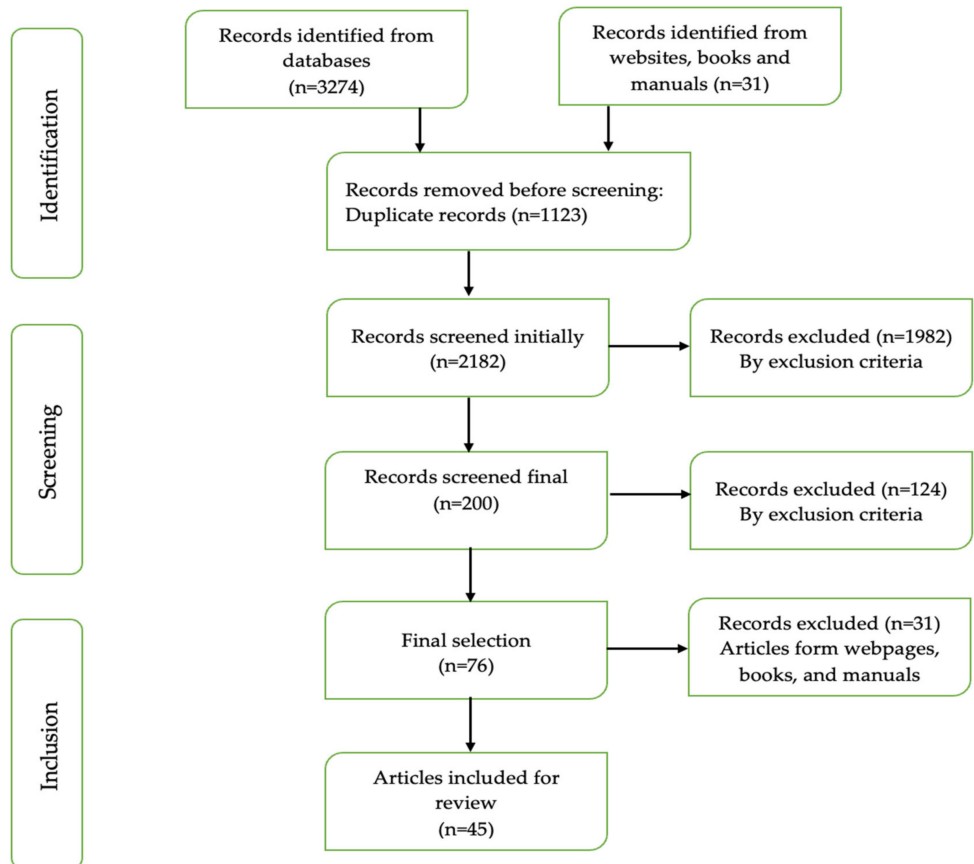

**Figure 1.** Flowchart for review process based on PRISMA guidelines.

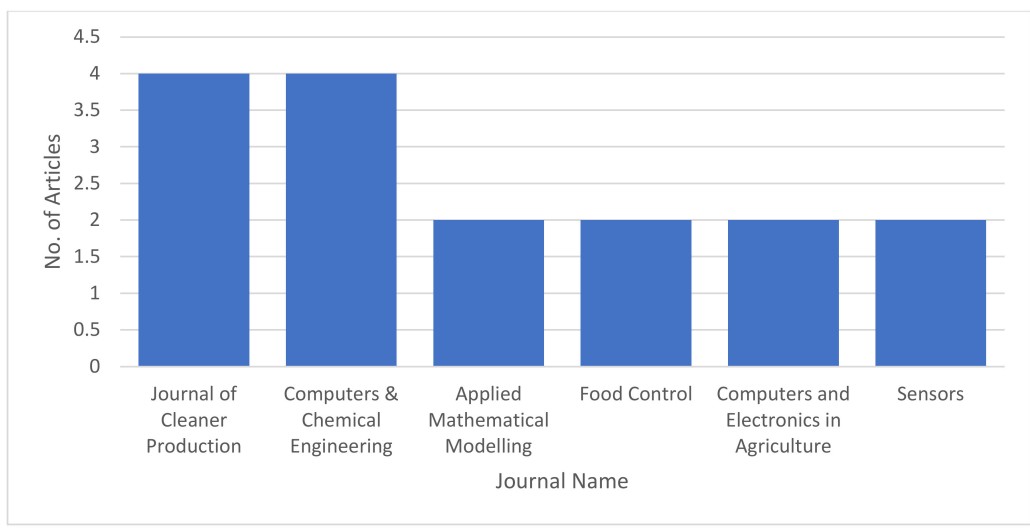

**Figure 2.** The number of articles per journal (if no. of articles $\geq$ 2).

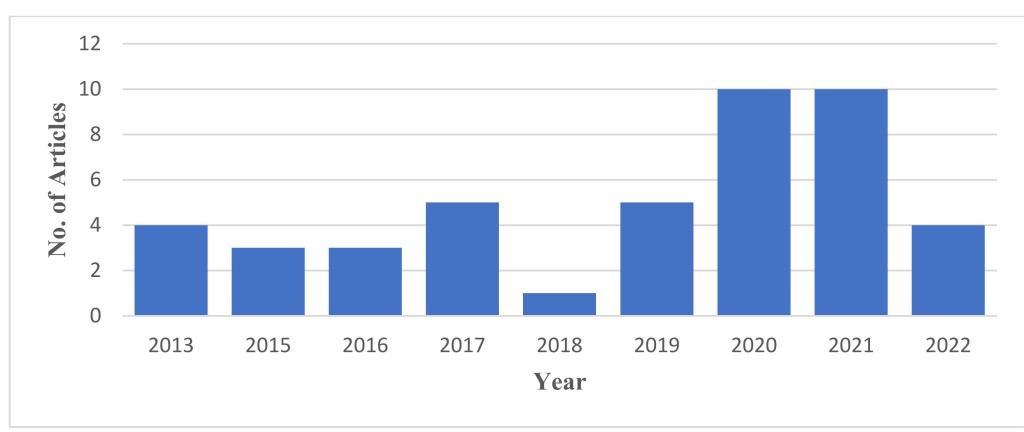

**Figure 3.** Publications per year.

### 3.4. Statistical Analysis

The data from each article were analyzed using the research questions as a guide. Basic features, including the number of papers in each journal and the year of publication, were retrieved and interpreted. Each article's optimization methods were studied to address the research question. The advantages and disadvantages of technology & ML, and AI were examined for the second question. Section 3 contains the results of this literature review. The findings are based on a content analysis of a small number of publications from the literature, considering the conclusions of the content analysis's results.

Figure 3 indicates the growing interest in supply chain optimization based on artificial intelligence and machine learning techniques may cause more yearly publications.

### 4. Discussion

Based on the outcomes of the systematic review and content analysis, this part discusses the analysis of selected articles. Table 2 summarizes the optimization techniques used in different areas of the DSC.

**Table 2.** Optimization techniques applied in DSC.

| Reference | Area | | | | Technique Used | |
|:---:|:---:|:---:|:---:|:---:|:---:|:---:|
| | Production | Processing | Inventory | Distribution | Mathematical Modelling | AI/ML/IoT |
| [9] | ✔ | | | | ✔ | |
| [10] | | | | ✔ | ✔ | ✔ |
| [12] | ✔ | | | | | ✔ |
| [13] | ✔ | | | | | ✔ |
| [15] | ✔ | ✔ | ✔ | ✔ | | ✔ |
| [16] | | ✔ | | | ✔ | |
| [17] | ✔ | | ✔ | | ✔ | |
| [18] | ✔ | | | ✔ | ✔ | ✔ |
| [19] | | | ✔ | | | ✔ |
| [20] | | ✔ | | | ✔ | |
| [21] | ✔ | ✔ | ✔ | ✔ | | ✔ |
| [22] | ✔ | | | | ✔ | ✔ |
| [23] | ✔ | ✔ | ✔ | ✔ | ✔ | |

**Table 2.** *Cont.*

| Reference | Area | | | | Technique Used | |
|:---:|:---:|:---:|:---:|:---:|:---:|:---:|
| | Production | Processing | Inventory | Distribution | Mathematical Modelling | AI/ML/IoT |
| [24] | | | ✔ | | ✔ | |
| [25] | ✔ | | | | | |
| [26] | ✔ | | ✔ | ✔ | ✔ | |
| [27] | | | ✔ | | | ✔ |
| [28] | ✔ | | | | ✔ | |
| [29] | ✔ | | | ✔ | | |
| [30] | ✔ | ✔ | ✔ | ✔ | | ✔ |
| [31] | | ✔ | | | | |
| [32] | ✔ | | | | | ✔ |
| [33] | ✔ | | | | ✔ | ✔ |
| [34] | | | | ✔ | ✔ | |
| [35] | | | | ✔ | ✔ | |
| [36] | ✔ | | | ✔ | ✔ | |
| [37] | ✔ | | | ✔ | ✔ | |
| [38] | ✔ | | | | | ✔ |
| [40] | | ✔ | | | ✔ | |
| [41] | | | | ✔ | ✔ | |
| [42] | ✔ | | | | | ✔ |
| [43] | ✔ | | | | | ✔ |
| [49] | ✔ | ✔ | | | ✔ | |
| [50] | ✔ | | | | | ✔ |
| [51] | | | | | | |
| [52] | | ✔ | ✔ | | ✔ | ✔ |
| [53] | ✔ | | | | | ✔ |
| [54] | ✔ | | | | | ✔ |
| [55] | ✔ | | | | | |
| [56] | | ✔ | | | | |
| [57] | ✔ | ✔ | ✔ | ✔ | ✔ | |
| [58] | ✔ | ✔ | ✔ | ✔ | ✔ | ✔ |
| [59] | | ✔ | | | ✔ | |
| [60] | ✔ | ✔ | ✔ | ✔ | ✔ | |
| [61] | ✔ | | | | ✔ | |

The authors highlighted the optimization techniques used in various DSC areas in Table 2. The categories for optimization techniques are as follows: part 1 includes all techniques based on mathematical modeling, and part 2 consists of all techniques based on artificial intelligence, machine learning, and IoT. Table 2 demonstrates that researchers are now using AI and ML-based algorithms for supply chain optimization. However, only 15% of studies address the entire supply chain optimization, leaving a gap in the field. Most studies, nearly 67%, deal with production and planning optimization, and only 29% are reported on inventory management.

### 4.1. Technological Innovations in the Dairy Supply Chain

This section focuses on the many DSC support solutions that use ML and AI. The conventional DSC is being changed or optimized using various techniques [13]. According to the study, there have been 14 disruptive innovations in the DSC. Multiple stages, activities, and supply chain procedures have been made more efficient via these technologies [28,62]. There were 16 physical and computerized innovations that the writers of the publications we were involved in examined, as indicated in Table 3. These technologies include simulations and AI, as well as ML and IoT. These include RFID, IoT, blockchain, cloud technology, data analytics, and remote sensing.

#### 4.1.1. Supportive Technologies in the Dairy Supply Chain

1. Automatic calf feeders are designed to supply nutrients to calves in exact doses throughout the day, resulting in a healthier and more productive life for the calves. It can control the maturation of the calves. It gives dairy cows the freedom to eat whenever they like [50,53]. Calves' development and health may be tracked in real-time, which benefits farmers.
2. Milk yield recording systems provide information about individual animals. You can see how much milk is produced each time a cow is milked and the daily trends in milk production. Farmers may use this early detection information to discover changes in the animals' health and food consumption routines [43]. As a result, it aids caregivers in identifying problems that could otherwise go undetected and implementing solutions.
3. Automated milking systems reduce the time and effort required to milk cows. Regarding milking, the cows can make their schedule and explore the refuge without human interference [63]. Computer-aided equipment detects the bovine, cleans the udder, softly collects the milk, and ultimately releases the animal after complete milking.
4. Rumination activity collars and sensors aid in the early identification of behavioral and activity abnormalities that indicate the onset of any illness or infection. It is possible to follow certain bovine activities, such as the animal's readiness to breed or enter labor, using devices such as pedometers [31]. This allows the farmer to give timely help.
5. Blockchain Technology shows a dairy farm's ability to produce milk. The technology is an electronic ledger with unique QR codes on the dairy label. These codes may provide information on the cow feed, possible treatment administration, raising procedure, slaughter event, time and place of milk production, processing processes, added ingredients, product release date, etc. [64]. Consumers may scan the tags with their cell phones to follow the whole milk manufacturing process from start to finish. This method attempts to eliminate the market for fake milk products.
6. AI Machine learning employs optical and motion sensors to analyze milk quality, fat and protein levels, and the reproductive health of cows. These devices may accurately transmit a cow's walking, drinking, eating, laying down, or contemplating. Health concerns like mastitis or lameness may be detected 24 h before they become serious [65]. These sensors can detect sick animals, allowing farmers to treat them quickly and prevent disease transmission.

#### 4.1.2. Supportive Technologies Implemented in Different Phases of the Dairy Supply Chain Herd Management Software

Herd management technology is transforming the dairy sector. Transparency and data analysis are the cornerstones of good herd management software [66]. The top solutions on the market extract and show real-time data in an easy-to-use style. To get actual ROI, you need to gather meaningful data and use it wisely. SCR by Allflex livestock intelligence is the most acceptable method for gaining these advantages: data collection and analysis for advanced cattle monitoring systems; heat detection and health monitoring in real-time; milk measuring technology that makes milking more accessible and more efficient; daily

dashboards with rapid drill-down access to management tools, including reports, graphs, analytics, task lists, and lifetime cow card history.

Farm Management Software

Given the varying farm characteristics, software assistance is required to simplify and automate these activities [67]. Using Agrivi's farm management system, users may plan, monitor, and analyze all agricultural operations, including weather monitoring, financial data and documentation management, human resource management, and inventory management.

Food Safety Software

Standardized safety procedures encourage openness and retain accurate records at every supply chain and production stage. CONTROL-PRO is the most excellent option for food safety in dairy production. As a result, users are more equipped to deal with food safety issues. It has dynamic features, such as making a floor layout with key control points and additional preventative control points; finding trends and places in the food safety strategy; interdepartmental and management communication; automating alerts; setting controls; easy access to all samples, testing, and correcting data; validating results; easy report generation and data analysis; early identification and response based on data component monitoring, trends and quick access to records reducing the audit preparation burden.

Supplier Management Software

From record-keeping and communication to audit planning and assessment, robust software can help you manage supplier relationships. Food LogiQ is the leading solution in this area, allowing customers to create a community of supply chain partners, address problems directly with suppliers, and achieve full chain traceability [40,68]. Features include easy-to-use supplier documentation and template procedures to verify and consolidate needed data; dashboards to quickly identify vendors with expired documents, failed audits, or high-quality problems and process credit requests, reduce supplier interactions, and catch quality complaints.

**Table 3.** Technologies utilized in DSC.

| Processes | Techniques | Application | Reference |
|---|---|---|---|
| Livestock Management | Smart Monitoring, Smartphone application, RFID, ANN, Clustering, Deep Learning | Grassland monitoring, animal welfare, livestock production, and decision-making based on accurate data and evidence | [32,42,43,50–53] |
| Procurement and Transportation | Simulation, IoT Blockchain, ANN, Genetic Algorithm | Quality and safety management, Route planning, Cost optimization | [10,12,33,55] |
| Planning and Production | ML such as Bayesian network clustering, genetic algorithm, Forecasting, ANN, Decision tree, Regression, SVM | Cost and profit Optimization in the production system to reduce setup time and better demand sensing | [20,22,49] |
| Inventory Management | Smart Phone application, genetic algorithm | To predict daily demand and deal with inventory-related problems | [17,19,24,28] |
| Dairy management | Simulation, Data analytics, robust and heuristic optimization | To achieve sustainability goals, to manage the data better | [9,21,26,36] |
| Distribution | Blockchain, robust optimization, regression, clustering | To predict future demand | [18,25,41,59,61] |
| Entire supply chain | Simulation, Cost optimization, Environmental impacts, Transportation optimization | To maintain transparency and visibility | [13,23,29–31,60] |

### 4.2. Strategic Outcomes

Strategic outcomes are the expected advantages and implications of applying these techniques to DSCs. The Dairy Supply Chain's predicted benefits and consequences are discussed below. The strategic outcomes have three dimensions: economic, environmental, and social. The strategic outcomes lower costs, improve efficiency, boost transparency, and minimize complexity. For example, RFID technologies help the dairy business by providing managers with real-time data to make better decisions [69]. Reduced greenhouse gas emissions and optimized energy use are critical environmental objectives. One study found that integrating technology into the DSC increased employment possibilities and reduced unlawful behavior. Cost reduction: The result might be a reduction in expenses for all parties involved. An IoT-enabled integrated system may cut dairy processing expenses. Researchers employed simulation to lower supply chain expenses [64]. A simulation-based methodology was designed to discover the best inventory strategy for the dairy business.

Efficiency: Supportive technologies have improved the efficiency of the DSC. Technology might improve dairy efficiency. RFID-based prototypes and tracking systems can boost economic efficiency. They deliver real-time data to stakeholders/managers to enhance supply chain management and operations. Transporting dairy products may be more efficient by adopting innovative technologies and limiting considerable truck weight [70,71]. Increase transparency: The DSC environment becomes more transparent for safe transactions between stakeholders. A traceability system based on RFID and blockchain helps collect data from farm to fork. This system aimed to earn consumer confidence by providing transparent supply chain information. Economic and technological advancement brought an untrustworthy environment. Transparency may be improved by combining numerous technologies and integrated systems [72,73]—complexity reduction: Less complicated than the usual DSC. Two studies focused on leveraging technology to simplify the DSC. An artificial neural network is easier and faster for researchers in the dairy industry simulations. The outcome reveals that the ML-based strategy facilitates decision-making [74]. However, few researchers have employed ML in the dairy business.

Legislative approval may be difficult because certain nations' governments and policies limit certain technologies. Governments did not provide any beneficial guidelines for the use of blockchain technology. Cybersecurity is a concern. The supply chain collects massive data from many digital devices. Various privacy and cyber and data security challenges arise due to the DSC's high degree of openness and interoperability [75]. Enhanced security, including anti-virus technology and software, is essential for expansion.

### 4.3. Future Research Directions

Even though various optimization techniques have addressed a wide range of problems over the past years in DSC, many issues still need to be resolved. This section describes the areas that can be addressed in the future. The increasing applications of emerging techniques based on AI and ML offer numerous opportunities for research into these gaps in DSC. Almost every study was focused on logistics process optimization, such as inventory, warehouse and transportation management [76]. Based on the findings, only 15% of studies addressed complete supply chain optimization. But with rising consumer demands, the industry requires an integrated optimization framework. Although segmented optimization is effective but rapid growth in demands necessarily involves the implementation of appropriate infrastructure. The DSC has not implemented AI- and ML-based techniques properly. Future studies might benefit the whole supply chain and create more jobs with technological implementation [42].

In addition, these technologies promote real-time data, interoperability, and integration amongst all partners in the supply chain. Supportive technologies facilitate the DSC. For the DSC, ML and new technologies are complicated, but proper implementation can improve supply chain operations [31]. However, certain technologies are seldom employed in practice or economically, particularly by SMEs. The key issue is adopting and adapting new technology. One factor for restricted technology adoption is high implementation costs.

Substantial investment challenges still exist in the DSC in digital technologies, the latest equipment, network infrastructure, and decision support systems. Most infrastructural facilities and cutting-edge technologies require government R&D funding. Future technological advancements may reduce the cost of adopting new technologies and discovering less costly materials [64]. Governments and organizations must assist dairy sector technology research and development to preserve assets, risk management and evaluation. Based on the findings and analysis, these are some key points that can be the area of research in the future:

- Integrated supply chain optimization;
- Real-time optimization for the routing problem;
- Real-time optimization for waste management in the supply chain;
- Real-time monitoring for information flow in the supply chain;
- Integrated optimization system based on AI and ML for the entire supply chain;
- Development of integrated frameworks for complete supply chain optimization;
- Implementation of optimizations models based on ML algorithms;
- The challenges and benefits of implementing optimization techniques can be explored;
- Factors affecting the adoption of digital techniques for DSC can be an area of research in future;
- Blockchain and IoT sensor-based real-time optimization in DSC.

Findings indicate that technological advancement is necessary for DSC, especially in real-time monitoring and integrated supply chain optimization.

## 5. Conclusions

Enhanced transparency, reduced cost, complexity and energy usage, and enhanced efficiency and employment possibilities are some implications of developing techniques based on ML and AI in the DSC. This review demonstrates that the potential of optimization techniques still needs a focus in the DSC. A brief overview of optimization techniques and how these techniques can enhance operations in the DSC is new to dairy. As per Table 2, optimization techniques are applied most frequently in the DSC production and planning phases, approx. 67% of articles, followed by distribution at 38%, processing at 33%, and inventory management at 29%. Also, around 15% of the articles covered the entire supply chain optimization. Also, mathematical modeling-based theoretical techniques are still dominant in the DSC compared to AI and ML-based algorithms, with 56% of studies reporting such techniques and 44% reporting those based on AI and ML. Thus, this review will help stakeholders understand current practices, their application in the DSC, and the advantages of ML and AI for the DSC. This review has some limitations too. The search criteria excluded some research in other languages. Also, some themes may be explored, including identifying system and network infrastructure risks in the DSC, Assessing ML and AI's suitability for the dairy industry on a small scale, and challenges to new technology adoption.

**Author Contributions:** Conceptualization, Formal analysis, and Writing—initial draft preparation and editing: M.M.; Review and Supervision: V.K.G., R.S.M., V.D. and M.Y. All authors have read and agreed to the published version of the manuscript.

**Funding:** This research received no external funding.

**Acknowledgments:** We acknowledge the Department of Basic and Applied Science and Department of Food Engineering, NIFTEM Kundli, for supporting review and analytical work.

**Conflicts of Interest:** The authors declare no conflict of interest.

# Appendix A

**Table A1.** List of articles included for systematic review.

| No. | Title |
|---|---|
| 1 | Dynamic dairy facility location and supply chain planning under traffic congestion and demand uncertainty: A case study of Tehran |
| 2 | Integrated production scheduling and distribution planning in dairy supply chain by hybrid modelling |
| 3 | RFID and perishable inventory management with shelf-space and freshness dependent demand |
| 4 | Strategies to characterize fungal lipases for applications in medicine and dairy industry |
| 5 | Decision Support System, Based on the Paradigm of the Petri Nets, for the Design and Operation of a Dairy Plant |
| 6 | Multi-bucket optimization for integrated planning and scheduling in the perishable dairy supply chain |
| 7 | Quantitative models for supply chain management within dairy industry: A review and discussion |
| 8 | A food traceability framework for dairy and other low-margin products |
| 9 | Possibilistic programming approach for production and distribution problem in milk supply chain |
| 10 | Recent advances in spray drying relevant to the dairy industry: A comprehensive critical review |
| 11 | Implementation of Artificial Intelligence Policy in the Field of Livestock and Dairy Farm |
| 12 | Integrated production and distribution scheduling with a perishable product |
| 13 | Order Picking Process in Warehouse: Case Study of Dairy Industry in Croatia |
| 14 | Planning and scheduling of the make-and-pack dairy production under lifetime uncertainty |
| 15 | The effect of Sales and Operations Planning (S&OP) on supply chain's total performance: A case study in an Iranian dairy company |
| 16 | A comprehensive model of demand prediction based on hybrid artificial intelligence and metaheuristic algorithms: A case study in dairy industry |
| 17 | A bi-objective multi-echelon supply chain model with Pareto optimal points evaluation for perishable products under uncertainty |
| 18 | An integrated two-layer network model for designing a resilient, green, closed loop supply chain of perishable products under disruption |
| 19 | Heuristic method for robust optimization model for green closed-loop supply chain network design of perishable goods |
| 20 | Optimal Production Scheduling in the Dairy Industries |
| 21 | Proposing a new model for location—Routing problem of perishable raw material suppliers with using meta-heuristic algorithms |
| 22 | A stochastic approach for integrated production and distribution planning in dairy supply chains |
| 23 | An intelligent Edge-IoT platform for monitoring livestock and crops in a dairy farming scenario |
| 24 | Artificial intelligence applied to a robotic dairy farm to model milk productivity and quality based on cow data and daily environmental parameters |
| 25 | Energy cost assessment of a dairy industry wastewater treatment plant |
| 26 | IoT for Development of Smart Dairy Farming |
| 27 | Machine learning based fog computing assisted data-driven approach for early lameness detection in dairy cattle |
| 28 | Optimization of Sampling for Monitoring Chemicals in the Food Supply Chain Using a Risk-Based Approach: The Case of Aflatoxins and Dioxins in the Dutch Dairy Chain |
| 29 | Robust design and planning for a multi-mode multi-product supply network: A dairy industry case study |
| 30 | The design and planning of an integrated supply chain for perishable products under uncertainties: A case study in milk industry |
| 31 | Towards combining data prediction and internet of things to manage milk production on dairy cows |
| 32 | A comparison of analytical test methods in dairy processing |

**Table A1.** *Cont.*

| No. | Title |
| --- | --- |
| 33 | A robust multi-objective optimization model for inventory and production management with environmental and social consideration: A real case of dairy industry |
| 34 | Biometric physiological responses from dairy cows measured by visible remote sensing are good predictors of milk productivity and quality through artificial intelligence |
| 35 | Future of dairy farming from the Dairy Brain perspective: Data integration, analytics, and applications |
| 36 | Implementation of technical and technological progress in dairy production |
| 37 | IoT-based Dairy Supply Chain—An Ontological Approach |
| 38 | Milk reception in a time-efficient manner: A case from the dairy processing plant |
| 39 | Multi-Product Multi Echelon Measurements of Perishable Supply Chain: Fuzzy Non-Linear Programming Approach |
| 40 | Robustness within the optimal economic poly generation system for a dairy industry |
| 41 | Sustainable closed-loop supply chain for dairy industry with robust and heuristic optimization |
| 42 | Environmental friendly route design for a milk collection problem: The case of an Indian dairy |
| 43 | Optimization of resource flows across the whole supply chain. Application to a case study in the dairy industry |
| 44 | Over 20 years of machine learning applications on dairy farms: A comprehensive mapping study |
| 45 | The role of livestock feed fertilization as an improvement of sustainability in the dairy sector |

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
