# Peer review of "Application of Optimization Techniques in the Dairy Supply Chain: A Systematic Review"

_logistics, 2022_

Round 1
Reviewer 1 Report
Green Application of Optimization Techniques in the Dairy Industry: A Systematic Review
Logistics MDPI
Deadline: 1 August 2022
Dear Editor,
Thank you for the opportunity to review this manuscript. Authors have been provided with a number of comments and recommendations designed to help them deliver better quality manuscript.
Dear Authors,
Thank you for conducting this systematic literature review, however I have serious concerns, mostly in reading sequence, that need to be resolved before I can recommend a decision:
General comments
· What do you mean by "green application"? The term "green application" seems unclear. Ensure that the title is straightforward and grammatically correct.
· Is the emphasis on the logistics process, the supply chain, or the industry? These are not the same terminology and have distinct theories attached to them. Make sure your title contains relevant phrases per your research objective. Your research objective/question seems to be focused on “dairy supply chain technologies”. You may have this in the title.
· The abstract need substantial revision. For instance, the opening sentences of the abstract are quite lengthy (7 lines - over half the abstract). It is also sounds confusing and incorrect. For example what do you mean by “the least and most” in the following sentence “As a result, the dairy industry needs to improve its operational efficiencies by implementing the latest and most effective optimization techniques.” Shouldn’t this be the most effective optimization techniques? I do not know in which situations “the least effective” could be favorable to improve operational efficiencies. I recommend using a professional proofreader, for the entire paper, to ensure that arguments are sound, and the material flows flawlessly. I also recommend that authors minimize repetition in their work. For instance, the final phrase of the abstract is redundant.
· The production of dairy products in India is given special attention in the introductory section, whereas the remaining sections of the manuscript focus on a more general and international setting. Which is the focus? The response to this inquiry need to be reflected across the entirety of the work.
· Authors did not provide any discussion and history of previous review papers on this topic. Authors need to justify why their review paper is needed in the text and providing and exhaustive list of all previous reviews on this topic either in the appendix or in the main text. Clearly explaining their novelty and contribution in the field. I strongly recommend authors to develop a new subtitle “Pervious literature reviews and research gap” after the introduction section. This future section is very critical and without this section it is impossible to recommend a decision on this manuscript. The following are some examples of reviews that have been done previously, but none of them have been referenced in your work:
o Feil, A. A., Schreiber, D., Haetinger, C., Haberkamp, Â. M., Kist, J. I., Rempel, C., ... & da Silva, G. R. (2020). Sustainability in the dairy industry: a systematic literature review. Environmental science and pollution research, 27(27), 33527-33542.
o Mor, R. S., Bhardwaj, A., & Singh, S. (2018). A structured-literature-review of the supply chain practices in dairy industry. Journal of Operations and Supply Chain Management, 11(1), 14-25.
· The authors state to have conducted a systematic literature review, yet the processes of a systematic literature review were not carefully followed. The keywords are either completely missing or reported in a very ambiguous style, despite the fact that this is an essential element of any systematic literature review. There is no reference to a well-established systematic literature review guideline. For example, the author could reference to PRISMA[1] or its modification[2] in supply chain domain, which I highly recommend following. I provide a recent sample of systematic literature review that you may adopt and revise your manuscript accordingly:
o Anjomshoae, A., Banomyong, R., Mohammed, F., & Kunz, N. (2022). A systematic review of humanitarian supply chains performance measurement literature from 2007 to 2021. International Journal of Disaster Risk Reduction, 102852.
· Authors should provide a complete list of reviewed articles in the appendix.
· Why did you chose 2018 as the beginning year for your review? What rationale supported this conclusion?
· In figure 2, there is no need to add journals with a single publication. Figure 2 and Figure 3 should present in solid color without shading. I recommend removing both figures and providing the results in a tabular style.
· Figure 4 is unnecessary. I can not see why the authors claim a theoretical framework based on generic language coupled to a standard graph. How does this figure advances readers understanding of the subject? The same is true for Figure 5. I suggest removing both figures.
· All figure and tables captions require correction and improved language. Check high-quality SLRs for figure captions, including the aforementioned systematic review.
· In the discussion section authors state “Based on aspects of technology, the framework has three component:”, but Figure 4 has 7 component. I found this very confusing to the readers. I recommend authors to streamline text with figure.
· What does the author intend by "framework" in relation to the previous comment? I do not understand what the author mean by “framework”. Is this a theoretical framework derived from the research findings? Or is this a framework that gives a classification of the topics they discovered in their review?
· Your framework is composed of three parts. How did you come up with those? The process in which they follow to develop a framework is completely missing. Author need to provide clear discussion on how their framework is developed based on their systematic review? This need to be developed in the next version of manuscript.
· There are too many bullet points in the discussion section. I strongly recommend author to move unnecessary wording and bullet points to the appendix.
· Overall, the discussion section needs a complete overhaul. In this section author need to provide thematic analysis, meaningful knowledge and insight, and clear and sound future research directions.
· Section 4 is very disjointed. It is unclear if this section is part of the results or the author's own elaboration. I propose merging this section with the discussion section and clearly linking it to the results.
· In the conclusion section, revise the sentence starting with “This research has significant drawbacks”. You could just simply say “this research has a number of limitations”.
Overall, I advise authors to devote considerable time to thoroughly addressing each of the aforementioned points in the next draft of their manuscript. A systematic literature review must adhere to a standardized process in order to produce trustworthy results. Authors must pay particular attention to the significance of their systematic review, discussion section, and future research recommendations. I strongly recommend that authors utilize a professional proofreader for their final manuscript revision.
[1] Moher, D., Shamseer, L., Clarke, M., Ghersi, D., Liberati, A., Petticrew, M., ... & Stewart, L. A. (2015). Preferred reporting items for systematic review and meta-analysis protocols (PRISMA-P) 2015 statement. Systematic reviews, 4(1), 1-9.
[2] Durach, C. F., Kembro, J., & Wieland, A. (2017). A new paradigm for systematic literature reviews in supply chain management. Journal of Supply Chain Management, 53(4), 67-85.
Author Response
Dear Sir/Madam,
Thank you very much for your valuable suggestions/comments on updating the work. Please see below the pointwise reply for your consideration.
|
No. |
Comment |
Response |
|
1 |
What do you mean by "green application"? The term "green application" seems unclear. Ensure that the title is straightforward and grammatically correct. |
In this paper, we have reviewed the applications of optimization techniques in the dairy supply chain and did not discussed about green application. |
|
2 |
Is the emphasis on the logistics process, the supply chain, or the industry? These are not the same terminology and have distinct theories attached to them. Make sure your title contains relevant phrases per your research objective. Your research objective/question seems to be focused on “dairy supply chain technologies”. You may have this in the title |
Thank you for pointing this out. Changes have been made, as suggested. |
|
3 |
The abstract need substantial revision. For instance, the opening sentences of the abstract are quite lengthy (7 lines - over half the abstract). It is also sounds confusing and incorrect. For example what do you mean by “the least and most” in the following sentence “As a result, the dairy industry needs to improve its operational efficiencies by implementing the latest and most effective optimization techniques.” Shouldn’t this be the most effective optimization techniques? I do not know in which situations “the least effective” could be favorable to improve operational efficiencies. I recommend using a professional proofreader, for the entire paper, to ensure that arguments are sound, and the material flows flawlessly. I also recommend that authors minimize repetition in their work. For instance, the final phrase of the abstract is redundant. |
Thanks for your valuable suggestion. Updated as suggested in lines 18-35. Abstract in updates as per guidelines and suggestions. |
|
4 |
The production of dairy products in India is given special attention in the introductory section, whereas the remaining sections of the manuscript focus on a more general and international setting. Which is the focus? The response to this inquiry need to be reflected across the entirety of the work. |
Thanks for your valuable suggestion. Updated as suggested in lines 71-88. |
|
5 |
Authors did not provide any discussion and history of previous review papers on this topic. Authors need to justify why their review paper is needed in the text and providing and exhaustive list of all previous reviews on this topic either in the appendix or in the main text. Clearly explaining their novelty and contribution in the field. I strongly recommend authors to develop a new subtitle “Pervious literature reviews and research gap” after the introduction section. This future section is very critical and without this section it is impossible to recommend a decision on this manuscript. The following are some examples of reviews that have been done previously, but none of them have been referenced in your work: · Feil, A. A., Schreiber, D., Haetinger, C., Haberkamp, Â. M., Kist, J. I., Rempel, C., ... & da Silva, G. R. (2020). Sustainability in the dairy industry: a systematic literature review. Environmental science and pollution research, 27(27), 33527-33542. o Mor, R. S., Bhardwaj, A., & Singh, S. (2018). A structured-literature-review of the supply chain practices in dairy industry. Journal of Operations and Supply Chain Management, 11(1), 14-25. |
The authors are thankful to the reviewer for his insightful comment. We updated the manuscript as suggested in lines 89-128 and 130-134. |
|
6 |
The authors state to have conducted a systematic literature review, yet the processes of a systematic literature review were not carefully followed. The keywords are either completely missing or reported in a very ambiguous style, despite the fact that this is an essential element of any systematic literature review. There is no reference to a well-established systematic literature review guideline. For example, the author could reference to PRISMA[1] or its modification[2] in supply chain domain, which I highly recommend following. I provide a recent sample of systematic literature review that you may adopt and revise your manuscript accordingly: · Anjomshoae, A., Banomyong, R., Mohammed, F., & Kunz, N. (2022). A systematic review of humanitarian supply chains performance measurement literature from 2007 to 2021. International Journal of Disaster Risk Reduction, 102852. |
Thanks, we have updated the manuscript as suggested. And updated manuscript as per PRISMA Guidelines in lines 129-205. |
|
7 |
Authors should provide a complete list of reviewed articles in the appendix. |
Appendix attached. |
|
8 |
Why did you chose 2018 as the beginning year for your review? What rationale supported this conclusion? |
Due to the limited number of relevant studies, we updated the years' span to 2013-2022. |
|
9 |
In figure 2, there is no need to add journals with a single publication. Figure 2 and Figure 3 should present in solid color without shading. I recommend removing both figures and providing the results in a tabular style. |
Updated. |
|
10 |
Figure 4 is unnecessary. I cannot see why the authors claim a theoretical framework based on generic language coupled to a standard graph. How does this figure advances readers understanding of the subject? The same is true for Figure 5. I suggest removing both figures. |
As suggested Figure 4 and 5 are removed. |
|
11 |
All figure and tables captions require correction and improved language. Check high-quality SLRs for figure captions, including the aforementioned systematic review. |
Updated as suggested. |
|
12 |
In the discussion section authors state “Based on aspects of technology, the framework has three component:”, but Figure 4 has 7 component. I found this very confusing to the readers. I recommend authors to streamline text with figure. |
We updated the discussion section and removed the figures. |
|
13 |
What does the author intend by "framework" in relation to the previous comment? I do not understand what the author mean by “framework”. Is this a theoretical framework derived from the research findings? Or is this a framework that gives a classification of the topics they discovered in their review? Your framework is composed of three parts. How did you come up with those? The process in which they follow to develop a framework is completely missing. Author need to provide clear discussion on how their framework is developed based on their systematic review? This need to be developed in the next version of manuscript. |
We updated the manuscript and summarized the findings in Table 2. Also, the systematic review process Flowchart is added to the manuscript. |
|
15 |
There are too many bullet points in the discussion section. I strongly recommend author to move unnecessary wording and bullet points to the appendix. Overall, the discussion section needs a complete overhaul. In this section, author need to provide thematic analysis, meaningful knowledge and insight, and clear and sound future research directions. |
Updated the discussion section, as suggested in lines 269-307. |
|
17 |
Section 4 is very disjointed. It is unclear if this section is part of the results or the author's own elaboration. I propose merging this section with the discussion section and clearly linking it to the results. |
Section 4 provides the challenges faced by the dairy supply chain when implementing novel techniques. |
|
18 |
In the conclusion section, revise the sentence starting with “This research has significant drawbacks”. You could just simply say “this research has a number of limitations”. |
Thank you for pointing out; we have updated the conclusion section. |
|
19 |
Overall, I advise authors to devote considerable time to thoroughly addressing each of the aforementioned points in the next draft of their manuscript. A systematic literature review must adhere to a standardized process in order to produce trustworthy results. Authors must pay particular attention to the significance of their systematic review, discussion section, and future research recommendations. I strongly recommend that authors utilize a professional proofreader for their final manuscript revision. [1] Moher, D., Shamseer, L., Clarke, M., Ghersi, D., Liberati, A., Petticrew, M., ... & Stewart, L. A. (2015). Preferred reporting items for systematic review and meta-analysis protocols (PRISMA-P) 2015 statement. Systematic reviews, 4(1), 1-9. [2] Durach, C. F., Kembro, J., & Wieland, A. (2017). A new paradigm for systematic literature reviews in supply chain management. Journal of Supply Chain Management, 53(4), 67-85. |
The authors are very thankful for these comments, which helped to update the manuscript more effectively. We have revised the manuscript in trace change mode.
|
Thanks and Regards

Reviewer 2 Report
The paper presents an interesting idea that could be versatile and useful to many research efforts. However, it needs a major revision before any feedback. For a review paper, it is essential to first understand the literature gap in this area. Why we need to know about this subject? Is there another review paper on this subject? If yes, what is the novelty of your research? In general, the paper starts with Machine Learning and optimization. But then it focuses on any software that is mentioned in other papers. What are the impacts? What are the optimizations? I would focus on how technology and AI affects the specific supply chains. Are other papers case-study based? If yes, then how they optimize supply chains (increasing transparency or economic efficiency by % etc in a specific supply chain)? The authors also need to re-write it as any technology, not machine learning. You mention enhancing transparency or supply chain optimization. But how? I believe this part of the paper needs work. I would expand this part of the paper on how actually technology and AI affect the supply chains (with concrete results). Also, in my point of view, the narrative behind this study needs improvement. In conclusion, the paper presents an interesting and important idea, which needs improvement in the results section and needs better communication.
Author Response
Dear Sir/Madam,
Good Day!
Thank your very much for your valuable suggestions/ comments for the further improvement of the work. Please see below the pointwise reply for your consideration.
|
S.N. |
Comments |
Response |
|
1 |
The paper presents an interesting idea that could be versatile and useful to many research efforts. However, it needs a major revision before any feedback. For a review paper, it is essential to first understand the literature gap in this area. Why we need to know about this subject? Is there another review paper on this subject? If yes, what is the novelty of your research? |
The authors are thankful to the reviewer for his insightful comment. We updated the manuscript as suggested, and it is highlighted. We discussed the application part of optimization techniques. |
|
2 |
In general, the paper starts with Machine Learning and optimization. But then it focuses on any software that is mentioned in other papers. What are the impacts? What are the optimizations? |
We updated the manuscript as suggested, impacts and optimization are given in line number 90-128 and table 2. |
|
3 |
I would focus on how technology and AI affects the specific supply chains. Are other papers case-study based? If yes, then how they optimize supply chains (increasing transparency or economic efficiency by % etc in a specific supply chain)? |
We explained the role of AI and ML in table 3 and % in line number 367-373. |
|
4 |
The authors also need to re-write it as any technology, not machine learning. You mention enhancing transparency or supply chain optimization. But how? I believe this part of the paper needs work. I would expand this part of the paper on how actually technology and AI affect the supply chains (with concrete results). |
Table 3 explains the technological role and the results are given in the conclusion section. |
Thanks and Regards
Reviewer 3 Report
The manuscript is well-written and the authors have provided various insights from the dairy industry focusing on optimization. It is a very interesting topic that fits the journal's aims and scope. The authors have provided a detailed presentation of their methodology and a very good discussion of the findings. There is ground for improving the conclusions section.
Author Response
Dear Sir/Madam,
Good Day!
Thank you very much for your valuable comments/suggestions for improving the work. Please see below the detailed reply for your consideration.
|
Comments |
Response |
|
The manuscript is well-written and the authors have provided various insights from the dairy industry focusing on optimization. It is a very interesting topic that fits the journal's aims and scope. The authors have provided a detailed presentation of their methodology and a very good discussion of the findings. There is ground for improving the conclusions section. |
The authors are thankful to the reviewer for his insightful comment. The conclusion section has been updated as suggested. |
Thanks and Regards
Round 2
Reviewer 1 Report
This manuscript has been updated following the first round of the review. Overall the manuscript has successfully done some very minor improvements based on previous comments. The manuscript still needs to go through another round of revision before it can be published. I consider that the modifications are minor and that certain mandatory improvements are still necessary. Major comments relate to 1) Justification for conducting a literature review, 2) Lack of rigor in conduct, 3) Poor command of English language 4) Lack of clear future research directions. My specific comments are as follows:
1. Line 146, 147, Does these statements look questions? Please reformat your RQs as questions rather than statements. My recommendation is to formulate research questions that begin with "how" and "why", but avoid "what" queries.
2. Please check my first review and address all the points thoroughly, including adding all the references that I provide in first revision round. Some bullet points have not been thoroughly addressed from the first revision round.
3. Figure 1: major points:
a. The statistics in the figure are wrong. Final selection should be 70 not 72?
b. Exclusion criteria is not clear. How did you exclude 1982 papers?
c. Put your exclusion/inclusion criteria in one table.
4. The outcome of a systematic literature review is "future research direction." Where are your future research directions? Create a table and list the future research directions. List a few bullet points outlining potential future research areas.
Overall, while the authors attempted to revise, the improvements are very marginal. I advise authors to strictly restrain from submitting a sloppy manuscript in the following round. Please 1) justify the need to conducting an SLR, 2) improve your arguments/language throughout the paper, 3) add research questions 4) add future research directions in a table format.
Best regards,
Author Response
We thank the reviewer (s) for sparing valuable time and providing comments/ suggestions for improving the manuscript. The detailed responses are attached herewith.
Thanks and Regards

Reviewer 2 Report
I believe the manuscript has improved sufficiently and it can be accepted in the present form.
Author Response
We thank the reviewer for sparing valuable time to comment/suggest further improvements in the manuscript. The responses are attached herewith.
Thanks and Regards

Round 3
Reviewer 1 Report
The authors have done sufficient improvement in this version of the manuscript. I have no further comments.